# Potential Tick Defense Associated with Skin and Hair Characteristics in Korean Water Deer (*Hydropotes inermis argyropus*)

**DOI:** 10.3390/ani14020185

**Published:** 2024-01-05

**Authors:** Sang-Joon Lee, Ki-Yoon Kim, Gyurae Kim, Subin Moon, Yung-Chul Park, Ho-Seong Cho, Yeonsu Oh

**Affiliations:** 1College of Veterinary Medicine and Institute of Veterinary Science, Kangwon National University, Chuncheon 24341, Republic of Korea; sjoon516@kangwon.ac.kr (S.-J.L.);; 2Division of Forest Science, College of Forest & Environmental Sciences, Kangwon National University, Chuncheon 24341, Republic of Korea; 3College of Veterinary Medicine, Chungbuk National University, Cheongju 28644, Republic of Korea; 4College of Veterinary Medicine and Bio-Safety Research Institute, Jeonbuk National University, Iksan 54596, Republic of Korea

**Keywords:** Korean water deer, skin and hair characteristics, comparative pathology, tick bites

## Abstract

**Simple Summary:**

Ticks, key vectors for multiple pathogens in veterinary medicine, lack the ability to fly. To feed on blood, they must climb onto a host’s hair coat from their natural habitat. Their life cycle is closely linked to wildlife, with the Cervidae family serving as primary hosts for adult ticks. In South Korea, the Korean water deer and roe deer are prominent members of the Cervidae family. Notably, the Korean water deer exhibit a lower rate of tick infestation compared to the roe deer. This study aims to understand the reasons behind this difference by conducting an in-depth comparative analysis of the skin and hair characteristics of both species. We found significant variations in certain factors that may hinder ticks from reaching their feeding sites on Korean water deer, potentially acting as a natural defense mechanism against ticks. To our knowledge, this research is the first to focus on comparing the skin and hair attributes of Korean water deer and roe deer in the context of tick bite prevention.

**Abstract:**

The Korean water deer (WD), a predominant wildlife species in South Korea, is listed as vulnerable by the IUCN Red List. Despite belonging to the same family, Cervidae, WD show significantly fewer adult ixodid tick infestations compared to roe deer (RD). Ticks, which cannot fly, engage in questing behavior in natural environments to latch onto hosts. They detect signals like body temperature and host skin chemicals to navigate through the hair coat to the preferred epidermis. In light of this, we performed an extensive comparative study of the skin tissue and hair characteristics of both deer species, focusing on elements contributing to the reduced tick bite incidence in WD. Remarkably, WD exhibited more prominent blood vessels, sebaceous glands, and sweat glands, which are crucial for skin barrier functions (*p* < 0.005). Moreover, WD had irregular scale patterns on their hair cuticles and possessed hair that was significantly stiffer and 2.83 times thicker than that of RD (*p* < 0.001). These characteristics potentially impede ticks from reaching the epidermis hair in WD and RD in the context of tick bite prevention. Further investigations in this area could enhance our understanding of tick–host dynamics and contribute to developing preventive measures against tick-borne diseases in other deer species.

## 1. Introduction

Ticks are recognized as significant vectors for a variety of protozoan, bacterial, and viral pathogens, impacting both public health and veterinary medicine more than any other group of blood-feeding arthropods [1]. Diseases such as severe fever with thrombocytopenia syndrome (SFTS), Lyme disease, tick-borne viral encephalitis, and Q fever are notable examples of tick-borne diseases affecting human and animals [1]. The recent global rise in tick-borne diseases is attributed to factors like climate change, land use modifications, expansion of host populations, and increased outdoor activities [2,3,4]. Ticks’ life cycles heavily depend on wildlife, which serve as both hosts and reservoirs, playing a pivotal role in the maintenance and transmission of tick-borne pathogens [5,6]. The Cervidae family, in particular, is a crucial host group for adult ticks [7].

Ticks commonly engage in a behavior known as “questing”, where they wait atop vegetation to latch onto a passing host [1]. They use sensory systems, such as Haller’s organs on their forelegs’ tarsi, to detect approaching hosts [1]. Once on a host, ticks seek optimal feeding sites, guided by stimuli like odor, body temperature, and humidity [8,9,10,11]. Factors like skin tissue properties, hair coat, and skin thickness can influence tick resistance. Furthermore, skin tissue with a hair coat can act as physical barriers that affect tick resistance, including the density of the fur coat and the skin thickness [12,13,14].

In the Republic of Korea, significant research on ticks, with over 143 articles published between 1966 and 2022, has identified prominent species in the Ixodidae and Argasidae families [15]. Notably, *Haemaphysalis longicornis*, *H. flava*, and *Ixodes nipponensis* are widespread. Among these, *H. longicornis* is the most prevalent (69.7%), followed by *H. flava* (22.2%), *I. persulcatus* (6.1%), *I. nipponensis* (1.8%), and *H. japonica* (0.2%) [16].

Korean water deer (WD; *Hydropotes inermis argyropus*), a member of the family Cervidae and a dominant species in the local ecosystem, are widely distributed across the entire Korean Peninsula, yet are classified as vulnerable by the International Union for Conservation of Nature (IUCN) [17]. Despite sharing several characteristics with other deer species, we found that the incidence of tick bites in WD is significantly lower than that in roe deer (RD; *Capreolus pygargus*) [18].

In this study, we explore the hypothesis that variations in skin tissue and hair coat between WD and RD may account for differences in tick bite incidence. We conducted a comparative analysis of the skin tissue and hair attributes of both deer species to understand these disparities better.

## 2. Materials and Methods

### 2.1. Tick Collection

During the active tick season, from July to October, we captured five individuals each of WD (Korean water deer) and RD (roe deer) from agricultural areas. We thoroughly inspected and collected ticks from the hair coats and body surfaces of these animals. The collected ticks were preserved in tick collection tubes for subsequent counting.

### 2.2. Sample Collection and Hair Analysis

Formalin-fixed skin tissues from two RD and two WD, selected from the captured animals, were collected from the Wildlife Research Lab at Kangwon National University. The selection was made based on the proximity of capture dates. Samples included tissues from four areas: head; neck; back; and axilla. From each tissue sample, ten guard hairs were randomly selected and extracted using forceps, leaving the remaining tissue for histopathological studies.

### 2.3. Hair Morphology Examination

We documented macroscopic features like hair type (straight or curled) and color. Hair shafts were then mounted on glass slides and secured with clear tape for measurements of diameter and length using a digital caliper and a light microscope equipped with CellSens standard 1.16 software (Olympus, Tokyo, Japan) [19]. The medulla structure was also examined. For cuticle analysis, we applied the gelatin casting method, leaving gelatin imprints of the hair scales for microscopic examination. The classification of cuticle and medulla patterns was based on established criteria [20,21]. Briefly, 20% pig native gelatin (Merck, Darmstadt, Germany) was prepared in boiling water. A thin film of gelatin was placed onto a clean glass slide. The hairs were placed superficially on the gelatin film and left overnight at room temperature. Subsequently, hairs were removed, leaving imprints of the scales on the gelatin cast. Images were acquired at ×400 using a BX53 light microscope (Olympus, Tokyo, Japan). Among the two subdivisions of guard hair, namely, the hair shield and the hair shaft, we selected the hair shaft, which is closer to the epidermis.

### 2.4. Histopathological Analysis

For each animal, we prepared four horizontal and four longitudinal sections from the skin tissues. The samples underwent a series of treatments, including dehydration, clearing in xylene, and embedding in paraffin blocks. Sections of 4 μm thickness were stained with hematoxylin and eosin (H&E) for histopathological comparison between the species, focusing on adnexa distribution, epidermal and dermal thickness, and follicle spacing. The analysis utilized NIH ImageJ software (version 1.54, Bethesda, MD, USA).

### 2.5. Statistical Analysis

Summary statistics were calculated for all groups to assess the overall quality of the data, including normality. The statistical relationship of hair measurements and histopathological results between RD and WD was determined with Student’s *t*-test, followed by the Mann–Whitney U test and Wilcoxon rank-sum test to calculate the post-power of the findings using IBM SPSS Statistics (version 26, Armonk, NY, USA). Differences were considered significant if *p* < 0.05.

## 3. Results

### 3.1. Variation in Tick Infestation between Roe Deer and Korean Water Deer

The study revealed a notably lower prevalence of adult ixodid tick infestations and blood feeding activities in WD compared to RD during the active tick season (July to October), with significant differences (*p* = 0.0079, 0.0317) (Table 1). The tick species identified include *Haemophysalis longicornis*, *H. flava*, *H. japonica*, and *Ixodes nipponensis* from the Ixodidae family. The observed variability in standard deviation and error, despite statistical significance, was attributed to the absence of tick infestations in August. The anomaly coincided with unusually dry weather and higher temperatures, averaging 1.5 °C above normal, marking it as the hottest year recorded [22].

### 3.2. Morphological Distrinction in Hair of Roe Deer and Korean Water Deer

Upon macroscopic examination, significant differences were noted in hair thickness and color between the species. RD featured yellow-brown coats, while WD had grayish coats. Distinct differences in hair cuticles, including scale position, scale margin structure, margin distance, and pattern, were observed between RD and WD [21]. WD had thicker and more irregular hair patterns, whereas RD displayed uniform scale margins (Figure 1A,B). In terms of medulla structure, both species showed similar characteristics, with other features being indistinguishable (Figure 1C,D).

### 3.3. Comparison of Hair Diameter and Length in Roe Deer and Korean Water Deer

We measured hair diameter and length from 20 hair shafts across four body parts (head, neck, back, and axilla) in both species (Table 2). RD exhibited significantly thinner hair diameters across all body parts compared to WD (*p* < 0.001), with WD hair being approximately 2.83 times thicker overall (*p* < 0.001). RD had shorter hair lengths in the head and back regions than WD (*p* < 0.001 and *p* = 0.020, respectively), but overall, hair lengths were not significantly different between the species (*p* = 0.437).

### 3.4. Skin Pathology Comparison between Roe Deer and Korean Water Deer

Our comparative pathological analysis focused on skin layers, blood vessels, and skin appendages, including sebaceous and sweat glands and hair follicles (Figure 2). Notable differences were observed in the sizes of skin appendages and hair follicle types, as well as the dermal blood vessel diameters. WD had larger skin appendages and simpler primary hair follicle pattens compared to RD’ compound primary with secondary follicles. Although the blood vessel distribution in the dermis was similar, WD had significantly larger blood vessel diameters. No notable differences were found in epidermal thickness. Quantitative measurements of sebaceous glands, sweat glands, and blood vessels showed significant enlargement in WD (*p* < 0.01) (Table 3), but no significant differences in epidermal thickness or distance between hair follicles’ midlines were found between the species (*p* = 0.403, 0.996).

## 4. Discussion

The premise of this investigation was to determine whether the lower incidence of adult tick bites in Korean water deer (*Hydropotes inermis argyropus*, WD) as compared to roe deer *(Capreolus pygargus*, RD) could be linked to distinct variations in their dermal or fur attributes. Our study approached this by examining the integumentary systems of both species, with a focus on characteristics such as hair and skin tissue.

Contrary to flying blood-feeding arthropods, ticks rely on direct contact with their hosts. This interaction is influenced by various phenotypic aspects like hair length, skin thickness, and coat texture. Prior studies, predominantly in cattle breeds, have established correlations between these characteristics and tick prevalence, noting that animals with finer, shorter coats and lighter colors generally demonstrate reduced tick burdens. Studies in cattle have shown a positive correlation between hair length and coat thickness with tick infestation in zebu, Holstein–Gyr [13], Ngun, Bonsmara, and Hereford cattle [23]. In other words, cattle with shorter hairs and smoother coats generally exhibit lower tick counts compared to those with longer hairs and woolier coats [13,24]. Also, lighter-colored coats have been linked to lower tick infection rates than darker-colored coats, suggesting that hair color can impact tick resistance [14,25].

Our research extends these findings to deer species, evaluating specific parameters such as hair color, cuticle structure, and follicle density [26]. In our analysis, WD and RD differed markedly in hair properties, including color, cuticular pattern, diameter, and density. While no significant variance in hair length was observed, WD’s hair was markedly thicker. Interestingly, hair density emerged as a crucial factor. WD’s hair spacing was significantly narrower, potentially hindering tick mobility and feeding site localization. The hair spacing in WD was over eight times narrower (26.64 μm) compared to that of RD (235.61 μm), which may cause adult ticks to spend more time finding feeding sites. Feeding sites differ throughout the ticks’ life stages; however, adult ticks primarily feed on the head and neck [7]. Furthermore, the microclimate created by these dense hairs could unfavorably affect tick survival [27]. Non-blood-feeding ticks can endure for about 40 h at 33% relative humidity [28], thus leading to prolonged exposure to the microclimate without rehydrating. The larger gaps between the hairs on RD, more than eight times wider than those of WD, could influence tick questing and penetration, potentially causing ticks to prefer RD as hosts.

Our comparative pathological examination of skin tissues revealed notable distinctions. WD possessed larger blood vessels and more prominent sweat and sebaceous glands. The role of skin appendages involves protection from the environment, temperature regulation, and maintenance of fluid balance, and the enhanced secretory output is attributed to the enlarged glands [29]. These physiological functions can influence the activity of blood-feeding arthropods [1,9,10], with WD primarily featuring simple primary follicles, while RD display compound primary with secondary follicle patterns. These factors, integral to skin health and homeostasis, could potentially alter the microenvironment, making it less hospitable for ticks. The difference in hair follicle structure and composition between the two species was also significant. WD predominantly featured stiffer primary hairs, while RD had softer, more pliable coats; such variations may impact the overall suitability of the host for ticks, considering factors like temperature and humidity regulation [13,14,23,24,25,30].

An emerging area of interest in vector biology is the role of the microbial community in attracting blood-feeding arthropods. The differential chemical signatures arising from microbial activity on the skin may influence host attractiveness to ticks [31]. Our study underscores the necessity of further research in this domain, particularly in understanding how host-specific microbiota and emitted semiochemicals, such as amino acids and fatty acids from the sweat and sebaceous glands, affect vector behavior [31].

In summary, this research provides a comprehensive comparison of the skin and fur characteristics of WD and RD, suggesting that the unique features of WD, including skin appendages, hair properties, and potential microbial influences, may confer a natural resistance to tick infestation. To the best of our knowledge, this is the first focused study comparing the skin and hair features of WD and RD in relation to tick bite prevention. This study not only advances our understanding of host–vector interactions in these species, but also lays the groundwork for exploring novel tick prevention strategies on other deer species and, potentially, in broader contexts.

## 5. Conclusions

This study aimed to elucidate the factors contributing to the lower tick infestation rates observed in Korean water deer (*Hydropotes inermis argyropus*, WD) compared to roe deer (*Capreolus pygargus*, RD), both belonging to the family Cervidae. Our hypothesis centered on the unique dermal and fur traits of WD. Through detailed comparative analysis, we identified several distinctive features in WD, including larger skin appendages, irregular cuticle patterns, increased hair thickness, and higher hair density. These findings illuminate the role of specific skin and fur characteristics in reducing tick prevalence in WD. The insights gained from this study are valuable for understanding the mechanisms of host defenses against tick infestation and could inform the development of preventive measures against tick-borne diseases in deer and, potentially, other species.

## Figures and Tables

**Figure 1 animals-14-00185-f001:**
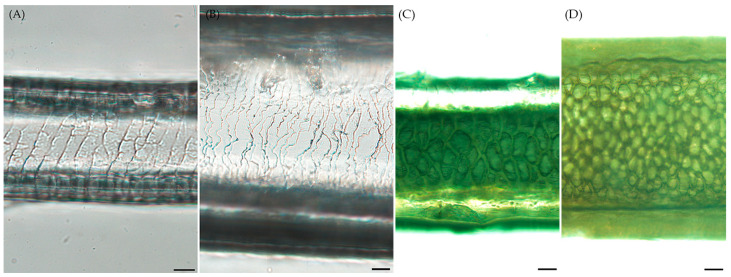
Hair cuticle and medulla characteristics. (**A**) Roe deer (*Capreolus pygargus*) hair cuticles exhibited transversal scale alignment, smooth scale edges, moderate distance between scales, and a consistent wave-like pattern. Scale bar = 20 μm. (**B**) Korean water deer (*Hydropotes inermis argyropus*) hair cuticles featured transversal scale alignment, slightly wavy scale edges, close proximity between scales, and an irregular wave pattern. Scale bar = 20 μm. (**C**) Roe deer (*Capreolus pygargus*) hair medullas were composed of multiple cells, a semi-filled lattice structure, a steady pattern, and scalloped edges. Scale bar = 50 μm. (**D**) Korean water deer (*Hydropotes inermis argyropus*) hair medullas were similar in structure with multiple cells, a semi-filled lattice, consistent pattern, and scalloped edges. Scale bar = 50 μm.

**Figure 2 animals-14-00185-f002:**
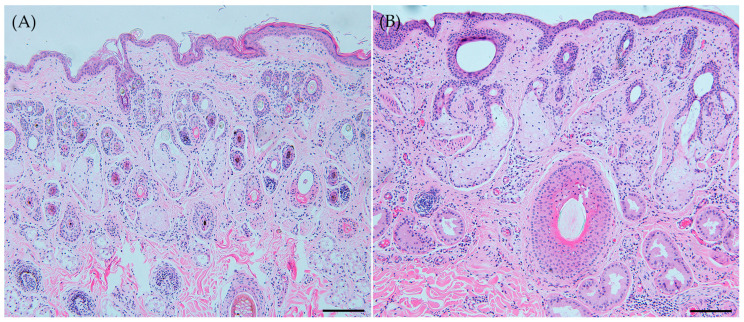
Cross-section of skin. (**A**) Roe deer (*Capreolus pygargus*) and (**B**) Korean water deer (*Hydropotes inermis argyropus*). In WD, the skin appendages were larger in size. The hair follicles in RD showed a compound primary with secondary follicle structure, more intricate compared to the basic primary follicle pattern seen in WD. WD had broader blood vessel diameters. The epidermal thickness did not show any notable variation. Scale bar = 200 μm. H&E staining.

**Table 1 animals-14-00185-t001:** Comparison of average counts of adult and blood-feeding ixodid ticks in Korean water deer and roe deer.

Species	No. of Adult Ixodid Ticks	No. of Blood-Feeding Ixodid Ticks
WD (*n* = 5)	25.2 ± 10.69	9.8 ± 4.18
RD (*n* = 5)	231.8 ± 81.81 **	85.2 ± 40.08 *

Values are presented as mean ± SE; *, significant difference compared to Korean water deer, *p* < 0.05; **, highly significant difference compared to Korean water deer, *p* < 0.01.

**Table 2 animals-14-00185-t002:** Analysis of hair diameter and length differences between roe deer and Korean water deer.

Parameter	Body Part	RD Hair (*n* = 20)	WD Hair (*n* = 20)
Diameter (μm) (Mean ± SD)	Head	136.49 ± 17.92 ***	216.65 ± 36.08
	Neck	172.45 ± 39.22 ***	436.63 ± 145.57
	Back	145.79 ± 17.74 ***	629.24 ± 26.93
	Axilla	118.88 ± 15.05 ***	326.86 ± 43.65
	Total	144.36 ± 3.47 ***	409.20 ± 167.56
Length (mm) (Mean ± SD)	Head	15.02 ± 3.04 ***	9.15 ± 1.49
	Neck	19.13 ± 3.55	20.56 ± 3.04
	Back	20.33 ± 3.50 *	24.03 ± 3.80
	Axilla	25.92 ± 4.56	25.59 ± 3.02
	Total	19.87 ± 5.24	19.98 ± 6.49

*, statistical significance between roe deer and Korean water deer, *p* < 0.05; ***, *p* < 0.001.

**Table 3 animals-14-00185-t003:** Comparison of skin appendage distribution in roe deer and Korean water deer.

Parameter	Tissue	RD	WD
Area (1000 μm^2^) (Mean ± SD)	Sebaceous gland (*n* = 24)	111.11 ± 20.12 ***	314.52 ± 22.84
	Blood vessel (*n* = 24)	9.25 ± 1.29 ***	42.26 ± 6.53
	Sweat gland (*n* = 24)	203.24 ± 33.73 **	310.49 ± 29.61
Distance (μm) (Mean ± SD)	Epidermis (*n* = 64)	62.23 ± 22.11	65.00 ± 12.48
	Primary hair follicle (*n* = 64)	390.08 ± 151.33	389.94 ± 119.05

**, statistical significance between roe deer and Korean water deer, *p* < 0.01; ***, *p* < 0.001.

## Data Availability

The data presented in this study are available in article.

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
