# Peer review of "Potential Tick Defense Associated with Skin and Hair Characteristics in Korean Water Deer (Hydropotes inermis argyropus)"

_animals, 2024, doi:10.3390/ani14020185_

Round 1
Reviewer 1 Report (Previous Reviewer 1)
Comments and Suggestions for Authors
The manuscript entitled “Potential tick defense associated with skin and hair characteristics in Korean water deer (Hydropotes inermis argyropus)” is interesting. I reviewed the manuscript submitted as animals-2650088. The manuscript has been significantly improved and the authors addressed all the concerns raised and completed the missing experiments like Variation in tick infestation between roe deer and Korean water deer. The looks good and may be accepted.
Comments on the Quality of English LanguageMinor editing is required.
Author Response
Thank you for the reviewer’s comment. We appreciate your insightful comments on animals-2650088.
Reviewer 2 Report (New Reviewer)
Comments and Suggestions for Authors
The authors mentioned that the incidence of tick infestation in the Korean water deer is significantly lower than that of the roe deer which seems interesting. The manuscript investigated the differences of skin tissues with hair coat between the two species and evaluated their potential functions in tick bites. The topic is interesting.
Lane 65 and 69-70. [Introduction] I recommend uniforming the common name, scientific name, and abbreviated form of animal species.
Lane 77 and 82. [Materials and Methods] The author used two animals for the tick counts and five individuals for the skin collection process. It is necessary to provide a description of whether the selection involved two out of the five individuals initially chosen or if an additional two animals were analyzed, and how the animal count was determined.
Lane 89 to 91. [Materials and Methods] Please check the font type.
Lane 181. [Discussion] The scientific name for Korean water deer should be italic.
Lane 198. [Discussion] Please check the typing error.
Author Response
Critique 1. Lane 65 and 69-70. [Introduction] I recommend uniforming the common name, scientific name, and abbreviated form of animal species.
Thank you for the reviewer’s comment. We uniformed the common name, scientific name, and abbreviated form of both species. The ‘roe deer (Capreolus pygargus, RD)’ has been changed into ‘roe deer (RD; Capreolus pygargus)’ in accordance with Korean water deer (WD; Hydropotes inermis argyropus).
Critique 2. Lane 77 and 82. [Materials and Methods] The author used two animals for the tick counts and five individuals for the skin collection process. It is necessary to provide a description of whether the selection involved two out of the five individuals initially chosen or if an additional two animals were analyzed, and how the animal count was determined.
Thank you for the reviewer’s comment. For tick counts, we captured five individuals of Korean water deer and five individuals of roe deer that intruded into agricultural areas during the active tick season from July to October. However, due to enhanced animal ethics considerations in the capture of wild animals, we were constrained to those sample size. We then selected two individuals from each species group for histopathologic analysis. The selection was made based on the proximity of captured dates. Those are reflected in the manuscript.
Critique 3. Lane 89 to 91. [Materials and Methods] Please check the font type.
Thank you for the reviewer’s comment. It seems that the font type might have changed in the middle. It was rechecked and unified into the Palatino Linotype reflected in the template.
Critique 4. Lane 181. [Discussion] The scientific name for Korean water deer should be italic.
Thank you for the reviewer’s comment. We changed the word into italic as the reviewer mentioned.
Critique 5. Lane 198. [Discussion] Please check the typing error.
Thank you for the reviewer’s comment. The typing error was revised.
This manuscript is a resubmission of an earlier submission. The following is a list of the peer review reports and author responses from that submission.
Round 1
Reviewer 1 Report
Comments and Suggestions for Authors
The authors mentioned that the incidence of tick infestation in the Korean water deer is significantly lower than that of the roe deer which seems interesting. The manuscript investigated the differences of skin tissues with hair coat between the two species and evaluated their potential functions in tick bites. The topic is interesting.
Introduction
The authors are requested to include the predominant tick species which infest family Cervidae of the particular region.
The authors mentioned that incidence of tick infestation in the Korean water deer is significantly lower than that of the roe deer which is based on their experience, is there any scientific reference? As the whole study is based on that fact, some published references need to be cited. Kindly mention.
Materials and methods
The authors mentioned that they analyzed skin tissues of two wild RDs and two wild WDs. How these animals were chosen? Randomly or based on tick susceptibility? Kindly explain…and only two samples seem low sample size.
Although the manuscript is interesting, the conclusion is not well supported by the findings. Susceptibility to tick infestation depends on several factors. The conclusion that ‘the lower tick bite frequency in water deer due to unique features of their skin tissues and fur’ is not very convincing.
In my opinion, the study is of very preliminary nature and not suitable for a prestigious journal like Animals.
Comments on the Quality of English LanguageIt is fine.
Reviewer 2 Report
Comments and Suggestions for Authors
I marked in the evaluation "accept but with major revisions." I explain why
A general problem I had while reading was not being able to clearly understand the meaning of some sentences, I recommend a revision of the English style to increase clarity.
I indicated English first because the next comments may be the result of misunderstanding due to lack of clarity while reading.
Personally, I appreciated the idea of looking for dermal and histological factors between two species belonging to the same family affected by ticks with different frequency, but I do not find in the manuscript a definite reference to this tendency of the parasite to prefer one of the two species, if it is an observation accrued by the authors during their exercise as researchers it could be indicated in this form.
It is not clear in the discussions (lines 184-185) how the "greater development of WD glands and vessels" could lead to a lower incidence of tick bite. Could an explanation of the hypothesis be written about this? I most likely observe the same lack in lines 189-190
Missing the citation: Pag.5, line 199, the brackets are blank
There are some papers that have already assessed the familiarity of ticks toward their hosts by mantle differences investigation. I list them below; it might be useful to use them in discussion.
1. Meltzer MI (1996) A possible explanation of the apparent breed-related resistance in cattle to Bont tick (Amblyomma hebraeum) infestations. Vet Parasitol 67:275–279
2. Foster A, Jackson A, D′Alteiro GL (2007) Skin diseases of South American Camelids. In Pract 29:216–222
3. Gasparin G, Miyata M, Coutinho LL (2007) Mapping of quantitative trait loci controlling tick [Rhipicephalus (Boophilus) microplus] resistance on bovine chromosomes 5, 7 and 14. Anim Genet 38:453–459
I personally cannot assess whether the method of investigation was the most appropriate to support the hypotheses made, but I think the result obtained is interesting if supported by further studies.
Comments on the Quality of English Language
i'm not english expert (i'm not the good writer of English) , I do not mean to sound disrespectful or offensive to the authors, and I apologize if I sounded improper.
For some sentences I had difficulty understanding the meaning or the subject. Perhaps some terms or vocabulary were misused. Some sentence constructions leads me to not understand the message from the authors.